# Deep Active Inference Agents for Delayed and Long-Horizon Environments

## Abstract

With the recent success of *world-model* agents—which extend the core idea of model-based reinforcement learning by learning a differentiable model for sample-efficient control across diverse tasks—*active inference* (AIF) offers a complementary, neuroscience-grounded paradigm that unifies perception, learning, and action within a single probabilistic framework powered by a generative model. Despite this promise, practical AIF agents still rely on accurate *immediate* predictions and exhaustive planning, a limitation that is exacerbated in *delayed* environments requiring planning over *long horizons*—tens to hundreds of steps. Moreover, most existing agents are evaluated on robotic or vision benchmarks which, while natural for biological agents, fall short of real-world industrial complexity. We address these limitations with a generative–policy architecture featuring (i) a *multi-step latent transition* that lets the generative model predict an entire horizon in a single look-ahead, (ii) an integrated policy network that enables the transition and receives gradients of the expected free energy, (iii) an alternating optimization scheme that updates model and policy from a replay buffer, and (iv) a single gradient step that plans over long horizons, eliminating exhaustive planning from the control loop. We evaluate our agent in an environment that mimics a realistic industrial scenario with delayed and long-horizon settings. The empirical results confirm the effectiveness of the proposed approach, demonstrating the coupled world-model with the AIF formalism yields an end-to-end probabilistic controller capable of effective decision making in delayed, long-horizon settings without handcrafted rewards or expensive planning.

## 1 Introduction

There has been significant progress in data-driven decision-making algorithms, particularly in reinforcement learning (RL), where agents learn policies through interaction with the environment and receive feedback [1]. Deep learning, in parallel, offers a powerful framework for extracting representations and patterns, while also enabling probabilistic modeling [2, 3], driving advancements in computer vision, natural language processing, biomedical applications, finance, and robotics. Deep RL merges these ideas—for example, by using neural function approximation in Deep Q-Networks (DQN), which achieved human-level performance on Atari games [4]. Model-based RL (MBRL) goes further by explicitly incorporating a model—either learned or provided—of the environment to guide learning and planning [5]. Similarly, the concept of world models centers on learning generative models of the environment to exploit representations and predictions of future outcomes, especially for decision-making [6]. This resonates with cognitive theories of the biological brain, which emphasize the role of internal generative models [7]. At a broader theoretical level, active inference (AIF), an emerging field in neuroscience, unifies perception, action, and learning in biological agents through the use of internal generative models [8, 9].

Submitted to 39th Conference on Neural Information Processing Systems (NeurIPS 2025). Do not distribute.

AIF is grounded in the free energy principle (FEP), which formulates neural inference and learning under uncertainty as minimization of *surprise* [10]. It provides a coherent mathematical framework that calibrates a probabilistic model governed by Bayesian inference, enabling both learning and goal-directed action directly from raw sensory inputs (i.e., *observations*) [9]. This can support the development of model-driven, adaptive agents that are trained end-to-end while offering uncertainty quantification and some interpretability [11, 12]. Similar to world models and model-based RL, AIF is powered by an internal model of the environment, which can help to capture dynamics and boost sample efficiency. Despite the potential of the AIF framework, its practical agents typically rely on accurate immediate predictions and extensive planning [12]. Such reliance can hinder performance, particularly in *delayed* environments, where the consequences of actions are not immediately observable—commonly framed in RL as *sparse rewards*, which exacerbates the credit-assignment problem [1]. Likewise, *long-horizon* tasks demand effective planning over extended temporal horizons, posing an additional challenge. These difficulties appear across diverse optimization tasks—such as manufacturing systems [11], robotics [13, 6, 14], and protein design [15, 16]—where the consequences become apparent only after many steps or upon completion of the entire process.

We explore how the potential of the AIF framework can be harnessed to build agents that remain effective in environments that are delayed and demanding long-horizon planning. Recent advances in deep generative modeling [17] have unlocked breakthroughs across diverse domains—such as AlphaFold's high-accuracy protein-structure predictions [18]. Because the generative model is the core of AIF, our objective is to extend its capacity and fidelity as the world model by predicting deep into the future. Concretely, we propose a generative model with an integrated policy network, trained end-to-end under the AIF formalism, allowing the model to produce long-horizon roll-outs and supply gradient signals to the policy during optimization. The summary of our contributions is as follows:

- We introduce an AIF-consistent generative–policy architecture that enables long-horizon predictions while providing differentiable signals to the policy.

- We derive a joint training algorithm that alternately updates the generative model and the policy network, and we show how the learned model can be leveraged during planning via gradient updates to the policy.

- We empirically demonstrate the concept's effectiveness in an industrial environment, highlighting its relevance to delayed and long-horizon scenarios.

The remainder of the paper is organized as follows: Section 2 reviews the formalism and planning strategies. Section 3 presents our proposed concept and agent architecture, while Section 4 details the experimental results. Finally, Section 5 concludes with implications and outlines future directions.

## 2 Background

Agents based on the world models concept extend the core idea of MBRL, learning a differentiable predictive model to facilitate policy optimization and planning via *imaginations* in the model [19, 6]. They create latent representations that capture spatial and temporal aspects to model dynamics and predict the future [19]. The architecture governing this dynamics—generative model—and how it is leveraged for policy and planning is foundational in this concept. Many designs resemble variational autoencoder [20] and are often augmented with Recurrent State-Space Models (RSSMs) to provide memory and help with credit assignment [21, 6, 14]. At the same time, RL methods such as actor–critic [1] are integrated with the model to optimize the policy [13, 6, 14], yielding sample-efficient agents that rely on imagination rather than extensive environment interaction.

AIF offers a complementary, neuroscience-grounded perspective that subsumes predictive coding that postulates that the brain minimizes prediction errors—relative to an internal generative model of the world—under uncertainty [22]. It casts the brain as a hierarchy that performs variational Bayesian inference continuously to suppress prediction error [9]. It was originally advanced to explain how organisms actively control and navigate their environments by iteratively updating beliefs and inferring actions from sensory observations [9]. AIF emphasizes the dependency of observations on actions [22]; accordingly, it posits that actions are chosen, while calibrating the model, to align with preferences and reduce uncertainty, thereby unifying perception, action, and learning [22]. The free-energy principle provides the mathematical bedrock for this framework

[23, 24], and a growing body of empirical work supports its biological plausibility [25]. AIF-based agents have been deployed in robotics, autonomous driving, and clinical decision support [26, 27, 28], demonstrating robust performance in uncertain, dynamic settings. In this work, we adopt the AIF formulation of Fountas et al. (2020) [12], which was extended in [29, 11] and has been shown to result in effective agents across different environments—such as visual and industrial tasks. We then review the planning strategies that can be coupled with this formalism.

## 2.1 Formalism

Within AIF, agents employ an integrated probabilistic framework consisting of an internal generative model [30] with inference mechanisms that allow them to represent and act upon the world. The framework assumes a Partially Observable Markov Decision Process [31, 30, 32], where an agent's interaction with its environment is formalized in terms of three random variables—observation, latent state, and action—denoted $(o_t, s_t, a_t)$ at time $t$. In contrast to RL, this formalism does not rely on explicit reward feedback from the environment; instead, the agent learns solely from the sequence of observations it receives. The agent's generative model, parameterized by $\theta$, is defined over trajectories as $P_\theta(o_{1:t}, s_{1:t}, a_{1:t-1})$ up to time $t$. The agent's behavior is driven by the imperative to minimize *surprise*, which is formulated as the negative log-evidence for the current observation, $-\log P_\theta(o_t)$ [12]. The agent approaches this imperative from two perspectives when interacting with the world, as follows [9, 12]:

1) Using the current observation, the agent calibrates its generative model by optimizing the parameters $\theta$ to yield more accurate predictions. Mathematically, this surprise can be expanded as follows [20]:

$$-\log P_\theta(o_t) \leq \mathbb{E}_{Q_\phi(s_t, a_t)} \left[ \log Q_\phi(s_t, a_t) - \log P_\theta(o_t, s_t, a_t) \right] , \quad (1)$$

which provides an upper bound, commonly known as the negative Evidence Lower Bound (ELBO) [33]. It is widely used as a loss function for training variational autoencoders [20]. In AIF, it corresponds to the Variational Free Energy (VFE), whose minimization reduces the surprise associated with predictions relative to actual observations [12, 34, 32].

2) Looking into the future, where the agent needs to plan actions, an estimate of the surprise of future predictions can be obtained. Considering a sequence of actions—or policy—denoted as $\pi$, for $\tau \geq t$, this corresponds to $-\log P(o_\tau | \theta, \pi)$, which can be estimated analogously to VFE [35]:

$$G(\pi, \tau) = \mathbb{E}_{P(o_\tau | s_\tau, \theta)} \mathbb{E}_{Q_\phi(s_\tau, \theta | \pi)} \left[ \log Q_\phi(s_\tau, \theta | \pi) - \log P(o_\tau, s_\tau, \theta | \pi) \right] . \quad (2)$$

This is known as the Expected Free Energy (EFE) in the framework, which quantifies the relative quality of policies—lower values correspond to better policies.

The EFE in Eq. 2 can be derived as a decomposition of distinct terms for time $\tau$, as follows [35, 12]:

$$G(\pi, \tau) = -\mathbb{E}_{\tilde{Q}} \left[ \log P(o_\tau | \pi) \right] \quad (3a)$$
$$+ \mathbb{E}_{\tilde{Q}} \left[ \log Q(s_\tau | \pi) - \log P(s_\tau | o_\tau, \pi) \right] \quad (3b)$$
$$+ \mathbb{E}_{\tilde{Q}} \left[ \log Q(\theta | s_\tau, \pi) - \log P(\theta | s_\tau, o_\tau, \pi) \right] , \quad (3c)$$

where $\tilde{Q} = Q(o_\tau, s_\tau, \theta | \pi)$ . Fountas et al. (2020) [12] rearranged this formulation with further use of sampling leading to a tractable estimate for the EFE that is both interpretable and easy to calculate [12]:

$$G(\pi, \tau) = -\mathbb{E}_{Q(\theta | \pi) Q(s_\tau | \theta, \pi) Q(o_\tau | s_\tau, \theta, \pi)} \left[ \log P(o_\tau | \pi) \right] \quad (4a)$$
$$+ \mathbb{E}_{Q(\theta | \pi)} \left[ \mathbb{E}_{Q(o_\tau | \theta, \pi)} H(s_\tau | o_\tau, \pi) - H(s_\tau | \pi) \right] \quad (4b)$$
$$+ \mathbb{E}_{Q(\theta | \pi) Q(s_\tau | \theta, \pi)} H(o_\tau | s_\tau, \theta, \pi) - \mathbb{E}_{Q(s_\tau | \pi)} H(o_\tau | s_\tau, \pi) . \quad (4c)$$

Conceptually, the contribution of each term in the EFE can be interpreted as follows [12]: Extrinsic value (Eq. 4a) — the expected *surprise*, which measures the mismatch between the outcomes predicted under policy $\pi$ and the agent's prior preferences over outcomes. This term is analogous to reward in RL, as it quantifies the misalignment between predicted and preferred outcomes. However, rather than maximizing cumulative reward, the agent minimizes surprise relative to preferred observations. State epistemic uncertainty (Eq. 4a) — mutual information between the agent's beliefs about states before and after obtaining new observations. This term incentivizes exploration of regions in the

environment that reduce uncertainty about latent states [12]. Parameter epistemic uncertainty (Eq. 4a) — the expected information gain about model parameters given new observations. This term also corresponds to active learning or curiosity [12], and reflects the role of model parameters $\theta$ in generating predictions. The last two terms capture distinct forms of epistemic uncertainty, providing an intrinsic drive for the agent to explore and refine its generative model. They play a role analogous to intrinsic rewards in RL that balance the exploration–exploitation trade-off. Similar information-seeking or curiosity signals underpin many successful RL algorithms—ranging from curiosity-driven bonuses [36, 37] to the entropy-regularized objective optimized by Soft Actor-Critic [38]—and have been shown to yield strong, sample-efficient agents.

## 2.2 Planning Strategy

Agents based on MBRL typically leverage their world model to *imagine* future trajectories before acting, trading extra computation for large gains in sample-efficiency and performance. Monte Carlo Tree Search (MCTS) [39, 40] is a notable search algorithm, which selectively explores promising trajectories in a restricted manner. Its effectiveness was highlighted in *AlphaGo Zero* [40] and later by *MuZero*, which folds a learned latent dynamics model directly into the search loop [41]. In the AIF concept, the agent's planning before taking actions is to minimize the EFE; mathematically, it corresponds to the negative accumulated EFE $G$ as follows:

$$P(\pi) = \sigma(-G(\pi)) = \sigma\left(-\sum_{\tau > t} G(\pi, \tau)\right) , \qquad (5)$$

where $\sigma(\cdot)$ represents the *Softmax* function. The agent simulates possible trajectories via roll-outs from its generative model, under policy $\pi$, to evaluate the EFE. However, calculating this any possible $\pi$ is infeasible as the policy space grows exponentially with the depth of planning. Fountas et al. (2020) [12] an auxiliary module along with the MCTS to alleviate this obstacle. They proposed a recognition module [42, 43, 44], parameterized with $\phi_a$ as follows: *Habit*, $Q_{\phi_a}(a_t)$, which approximates the posterior distribution over actions using the prior $P(a_t)$ that is returned from the MCTS [12]. This is similar to the fast and habitual decision-making in biological agents [45]. They used this module for fast expansions of the search tree during planning, followed by calculating the EFE of the leaf nodes and backpropagating over the trajectory. Iteratively, it results in a weighted tree with memory updates for the visited nodes. They also used the Kullback–Leibler divergence between the planner's policy and the habit provides as precision to modulate the latent state [12]. They also used the Kullback–Leibler divergence between the planner's policy and the habit provides as precision to modulate the latent state [12]. Another approach to enhance the planning is using a *hybrid horizon* [11], in which the short-sighted EFE terms—based on immediate next-step predictions—are augmented with an additional term during planning to account for longer horizons. Taheri Yeganeh et al. (2024) [11] employed a Q-value network, $Q_{\phi_a}(a_t)$, to represent the amortized inference of actions, trained in a model-free manner using extrinsic values. These terms were then combined in the planner as follows:

$$P(a_t) = \gamma \cdot Q_{\phi_a}(a_t) + (1 - \gamma) \cdot \sigma\left(-G(\pi)\right) , \qquad (6)$$

balancing long-horizon extrinsic value against short-horizon epistemic drive.

Modern world-model agents increasingly shift the look-ahead into latent space; PlaNet [21] uses cross-entropy method roll-outs inside a RSSM trained with *latent overshooting*, while the Dreamer family [13, 6] propagates analytic value gradients through hundreds of imagined trajectories, without a tree search. EfficientZero [46] blends AlphaZero-style MCTS with latent-space imagination, surpassing human Atari performance with only 100k frames. These approaches typically couple multi-step model roll-outs with an actor (policy) and often a critic (value) network that are queried during imagination. In each simulated step, the policy proposes the next action and the critic supplies a bootstrapped value, enabling efficient multi-step look-ahead without enumerating the full action tree. Instead of sequentially sampling actions and states, Taheri Yeganeh et al . [11] trained multi-step latent transitions, conditioned on repeated actions; during planning, a single transition predicts the outcome while keeping an action for a fixed number of time-steps. This way, the impact of actions over a long horizon is captured using repeated-action simulations. While it can be combined with MCTS, this approximation helps distinguish different actions based on the EFE in highly stochastic control tasks with a single look-ahead [11]. It is limited to discrete actions, cannot go beyond repeated actions, and still requires planning via EFE computation before every action.

## 3 Deep Active Inference Agent

From habit-integrated MCTS to hybrid-horizon and gradient-based latent imagination, state-of-the-art agents increasingly integrate policy learning with planning to capture the long-term effects essential for scalable and sample-efficient control. A prominent approach is latent imagination, notably used by Dreamer agents [6, 21, 13], which perform sequential rollouts in latent space using a RSSM. Besides its computational burden, this method risks accumulating errors as networks are repeatedly inferred and sampled. These models embed the policy network in the latent space by sampling actions along each latent-state trajectory, so policy optimization depends on a large number of samplings in the model's imaginations.

A simpler strategy assumes a generative model that *knows* the exact form of the policy function—in other words, the network parameters themselves. We can train such a model to generate a prediction deep into the horizon with a single look-ahead, once provided with the policy parameters governing interaction with the environment over that horizon. Thus, the EFE can be computed directly over the horizon, and its gradients can be backpropagated to minimize the EFE while still guiding the agent toward its intrinsic and extrinsic objectives. Given that the policy is optimized through the gradient steps of the EFE, this approach naturally scales to continuous action space rather than choosing discrete actions, as in earlier AIF-agent implementations[12]. Here, we adopt this AIF-consistent generative-policy modeling, without incorporating further mechanisms typically employed to further enhance world models or AIF agents.

### 3.1 Architecture

The agent comprises, at a minimum, a **policy network** that directly interacts with the environment and a **generative model** that is trained to optimize that policy. Conditioned on the policy, the generative model constitutes the core of AIF and can be instantiated with various architectures. In this work we adopt a generic—yet commonly used—autoencoder assembly [12] to instantiate the formalism of Sec. 2.1, which requires the tightly coupled modules illustrated in Fig. **??**. Leveraging amortization [20, 43, 47] to scale inference [12], the generative model is parameterized by two sets: $\theta = \{\theta_s, \theta_o\}$ for prior generation and $\phi = \{\phi_s\}$ for recognition. Accordingly, the **Encoder** $Q_{\phi_s}(s_t)$ performs amortized inference by mapping the currently sampled observation $\tilde{o}_t$ to a posterior distribution over the latent state $s_t$ [48]. The key difference here is that, rather than sampling actions inside the latent dynamics, we incorporate a policy function—or **Actor**—$Q_{\phi_a}(a_t \,|\, \tilde{o}_t)$, which itself infers a distribution over actions with parameters $\phi_a$. We therefore introduce an explicit representation for the function itself with the mapping $\Pi : \mathcal{Q}_{\phi_a} \to \hat{\pi}$, resulting in $\hat{\pi}(\phi_a)$. This approach is common in neural implicit representations [49]; recent work has moreover demonstrated that neural functions with diverse computational graphs can be embedded efficiently [50]. Conditioned on the actor, the **Transition**, $P_{\theta_s}(s_{t+1} \,|\, \tilde{s}_t, \hat{\pi})$, *overshoots* the latent dynamics up to a planning horizon $H$, producing a distribution for $s_{t+H}$ given the sampled latent state at time $t$, while the actor–denoted by $\phi_a$–is assumed fixed throughout the horizon. Finally, the **Decoder** $P_{\theta_o}(o_{t+H} \,|\, \tilde{s}_{t+H})$ converts the predicted latent state back into a distribution over future observations.

Each of the three modules in the generative model is realized by a neural network that outputs the parameters of a diagonal multivariate Gaussian, thereby approximating a pre-selected likelihood family. They can be trained end-to-end by minimizing the VFE (Eq. 1), whereas the actor is optimized—using predictions from the calibrated model—by minimizing the EFE (Eq. 4). In this way, the agent unifies the two free-energy paradigms derived in the formalism. Aside from the actor and transition, which account for latent dynamics with a single look-ahead, the architecture resembles a variational autoencoder (VAE) [20]; nevertheless, other generative mechanisms, such as diffusion or memory-based RSSM models, can be extended to support the same objective.

### 3.2 Policy Optimization

We propose a concise yet effective formulation for embedding the actor within the generative model so that it serves as a planner that minimizes the EFE via gradient descent. Conditioned on a fixed policy $\hat{\pi}(\phi_a)$, the model generates the prediction distribution $P_\theta(o_{t+H} | \phi_a)$, from which we compute the EFE, denoted as the function $G_\theta(\tilde{o}, \phi_a)$. Policy optimization then proceeds by updating the actor parameters according to the gradient $\nabla_{\phi_a} G_\theta(\tilde{o}, \phi_a)$. Most world-model agents introduce stochasticity by sampling actions during imagination, which promotes exploration—typically aided by auxiliary

terms during the policy gradient. This results in a Monte Carlo estimation of the policy across imagined trajectories, which is then differentiated based on the return [13]. In contrast, our approach assumes the exact form of the policy is integrated into the dynamics, and exploration is driven by the AIF formalism based on the generative model.

To effectively estimate the different components of the EFE in Eq. 4, Fountas et al. (2020) [12] employed multiple levels of Monte Carlo sampling. While their original formulation incorporated sampled actions over multi-step horizons, the same structure and sampling scheme remain beneficial when using an integrated actor with deep temporal overshooting. Similarly, we adopt ancestral sampling to generate the prediction $P_\theta(o_{t+H} \mid \phi_a)$ and leverage dropout [51] in the networks. It's coupled with further sampling from the latent distributions to compute the entropies necessary for calculating the EFE terms. Crucially, under the AIF framework, agents need a prior preference over predictions to guide behavior—this is formalized through the extrinsic value (i.e., Eq. 4a). Accordingly, we define an analytical mapping that transforms the prediction distribution into a continuous preference spectrum, $\Psi : P_\theta(o_\tau) \rightarrow [0, 1]$.

Unlike RL, which relies on a monotonic return value based on accumulated rewards, this formulation allows the agent to express more general and nuanced forms of preference. In practice, designing a suitable reward function for RL agents remains a difficult task, often resulting in sparse or hand-crafted signals that can be costly to design and compute. The flexibility in preference, however, introduces challenges—particularly when agents have complex preference-space and must act with short-sighted EFE approximations. Our approach, by optimizing planning through deep temporal prediction, mitigates this issue and enables longer-term evaluation of the extrinsic value.

### 3.2.1 Training & Planning

During training, the generative model gradually learns how different actor parameters $\phi_a$ affect the dynamics, and during policy optimization, this learned dynamics is then used to differentiate the actor toward lower EFE or surprise. Critical for effective policy learning is the accuracy of the world model, which forms the foundation of AIF [23, 9, 12] and predictive coding [22]. To improve model training, we introduce experience replay [4] using a memory buffer $\mathcal{M}$, from which we sample batches of experiences, while ensuring that each batch includes the most recent transition. We compute the VFE in Eq. 1 for these experiences to train the model with $\beta$-regularization. With the updated model, we differentiate the EFE over a batch of observations—including previous and current ones—within imagined trajectories of length $H$, training the actor similarly to world-model methods [13, 6, 19]. This results in a joint training algorithm 1 that alternates between updating the generative model and the policy, using the model to guide planning via policy gradients. This approach, policy learning—rather than explicit action planning—relaxes the bounded-sight constraint of EFE, as the policy is iteratively trained across diverse scenarios within the planning horizon, and its effective *sight* extends beyond the nominal horizon $H$. Recent work on AIF-based agents has also emphasized the advantages of integrating a policy network with the EFE objective [14]. After training concludes and the agent's model is fixed, the agent can still leverage its model for planning. Specifically, EFE-based gradient updates can be applied at the observation level once every $H$ steps, effectively fine-tuning the policy for the immediate horizon.

## 4 Experiments

Most existing AIF agents have shown effectiveness across a range of tasks typically performed by biological agents, such as humans and animals. These tasks often involve image-based observations [14]. For example, Fountas et al. (2020) [12] evaluated their agent on Dynamic dSprites [52] and Animal-AI [53], which biological agents can perform with relative ease. AIF has also been successfully applied in robotics [54, 29], including object manipulation [14, 27], aligning with behaviors humans naturally perform. This effectiveness is largely attributed to AIF's grounding in theories of decision-making in biological brains [9]. However, applying AIF to more complex domains—such as industrial system control—poses significant challenges. Even humans may struggle to design effective policies in these settings. Such environments often exhibit high stochasticity, where short observation trajectories are dominated by noise, making it difficult to optimize free energy for learning and action selection. This issue is less pronounced in world model agents, which often use memory-based (e.g., recurrent) architectures [13, 6]. Moreover, realistic environments frequently combine discrete and continuous observation modalities, complicating generative and

**Algorithm 1** Deep AIF Agent Training (per epoch)

| | |
|---|---|
| 1: Initialize $\theta = \{\theta_s, \theta_o\}$, $\phi = \{\phi_s, \phi_a\}$, $\mathcal{M}$ | **Agent components:** |
| 2: Randomly initialize $E$ | Model: |
| 3: **for** $n = 1, 2, ..., N$ **do** | Encoder $Q_{\phi_s}$. |
| 4: $\quad \hat{\pi}_t \leftarrow \Pi(\mathcal{Q}_{\phi_a})$ | Transition $P_{\theta_s}$. |
| 5: $\quad$ **for** $\tau = t + 1, t + 2, ..., t + H$ **do** | Decoder $P_{\theta_o}$. |
| 6: $\quad\quad$ Sample a new observation $\tilde{o}_\tau$ from $E$ | Actor $Q_{\phi_a}$. |
| 7: $\quad\quad$ Apply $\tilde{a}_\tau \sim Q_{\phi_a}(a_\tau|\tilde{o}_\tau)$ to $E$ | Actor mapping $\Pi$. |
| 8: $\quad\quad$ Sample a new observation $\tilde{o}_{\tau+1}$ from $E$ | Preference mapping $\Psi$. |
| 9: $\quad \mathcal{M} \leftarrow \mathcal{M} \cup \{(\tilde{o}_t, \hat{\pi}_t, \tilde{o}_{t+H})\}$ | |
| 10: $\quad \{(\tilde{o}_{t'}, \hat{\pi}_{t'}, \tilde{o}_{t'+H})\} \sim \mathcal{M}$ | **Other components:** |
| 11: $\quad$ **for** $t' = 1, 2, ..., B_m$ **do** | Environment $E$. |
| 12: $\quad\quad$ **run** Model$(\tilde{o}_{t'}, \hat{\pi}_{t'}, \tilde{o}_{t'+H})$ | Memory buffer $\mathcal{M}$. |
| 13: $\quad\quad \mathcal{L}_s \leftarrow \mathcal{L}_s + D_{\text{KL}}\left[Q_{\phi_s}(s_{t'+H}) \| \mathcal{N}(\mu, \sigma^2)\right]$ | |
| 14: $\quad\quad \mathcal{L}_o \leftarrow \mathcal{L}_o - \mathbb{E}_{Q(s_{t'+H})}\left[\log P_{\theta_o}(o_{t'+H}|\tilde{s}_{t'+H})\right]$ | **Hyperparameters:** |
| 15: $\quad\quad \mathcal{L}_o \leftarrow \mathcal{L}_o + \beta * D_{\text{KL}}\left[Q_{\phi_s}(s_{t'+H}) \| \mathcal{N}(\tilde{\mu}, \tilde{\sigma}^2)\right]$ | Iterations $N$. |
| 16: $\quad \theta_s \leftarrow \theta_s - \xi \nabla_{\theta_s} \mathbb{E}\left[\mathcal{L}_s(\theta_s)\right]$ | Beta $\beta$. |
| 17: $\quad \phi_s \leftarrow \phi_s - \gamma \nabla_{\phi_s} \mathbb{E}\left[\mathcal{L}_s(\phi_o)\right]$ | Horizon $H$. |
| 18: $\quad \theta_o \leftarrow \theta_o - \eta \nabla_{\theta_o} \mathbb{E}\left[\mathcal{L}_o(\theta_o)\right]$ | Batch size $B_m$, $B_a$. |
| 19: $\quad$ **for** $\tau = 1, 2, ..., B_a$ **do** | Sample size $S$. |
| 20: $\quad\quad \{\tilde{o}_\tau\} \sim \mathcal{M}$ | Learning rate $\xi, \gamma, \eta, \alpha$. |
| 21: $\quad\quad$ Compute $Q_{\phi_s}(s_\tau)$ using $\tilde{o}_\tau$ | |
| 22: $\quad\quad$ Sample $\tilde{s}_\tau \sim Q_{\phi_s}(s_\tau)$ | **Run** Model$(\tilde{o}_i, \hat{\pi}, \tilde{o}_{i+H})$: |
| 23: $\quad\quad$ **for** $s = 1, 2, ..., S$ **do** | Compute $Q_{\phi_s}(s_i)$ using $\tilde{o}_i$ |
| 24: $\quad\quad\quad$ Compute $\mu, \sigma$ from $P_{\theta_s}(s_{\tau+H}|\tilde{s}_\tau, \hat{\pi}_t)$ | Sample $\tilde{s}_i \sim Q_{\phi_s}(s_i)$ |
| 25: $\quad\quad\quad$ Sample $\tilde{s}_{\tau+H} \sim \mathcal{N}(\mu, \sigma^2)$ | Compute $\mu, \sigma \leftarrow P_{\theta_s}(s_{i+H}|\tilde{s}_i, \hat{\pi})$ |
| 26: $\quad\quad\quad$ Compute $P_{\theta_o}(o_{\tau+H}|\tilde{s}_{\tau+H})$ | Compute $Q_{\phi_s}(\tilde{s}_{i+H})$ using $\tilde{o}_{i+H}$ |
| 27: $\quad\quad\quad$ Compute $Q_{\phi_s}(\tilde{s}_{\tau+H})$ using $\tilde{o}_{\tau+H}$ | Compute $\mu', \sigma' \leftarrow Q_{\phi_s}(\tilde{s}_{i+H})$ |
| 28: $\quad\quad\quad$ Compute $\mu', \sigma' \leftarrow Q_{\phi_s}(\tilde{s}_{\tau+H})$ | Sample $\tilde{s}_{i+H} \sim \mathcal{N}(\mu, \sigma^2)$ |
| 29: $\quad\quad\quad G \leftarrow G + \Phi(P_{\theta_o}(o_{\tau+H}|\tilde{s}_{\tau+H}))$ | Compute $P_{\theta_o}(o_{i+H}|\tilde{s}_{i+H})$ |
| 30: $\quad\quad\quad G \leftarrow G + [H(\mu', \sigma') - H(\mu, \sigma)]$ | |
| 31: $\quad\quad$ **for** $s = 1, 2, ..., S$ **do** | |
| 32: $\quad\quad\quad$ Sample $\tilde{s}_{\tau+H} \sim P_{\theta_s}(s_{\tau+H}|\tilde{s}_\tau, \hat{\pi}_\tau)$ ▷ *Re-computed with dropout.* | |
| 33: $\quad\quad\quad$ Compute $\mu'', \sigma'' \leftarrow P_{\theta_o}(o_{\tau+H}|\tilde{s}_{\tau+H})$ | |
| 34: $\quad\quad\quad$ Sample $\tilde{s}_{\tau+H} \sim \mathcal{N}(\mu, \sigma^2)$ | |
| 35: $\quad\quad\quad$ Compute $\mu''', \sigma''' \leftarrow P_{\theta_o}(o_{\tau+H}|\tilde{s}_{\tau+H})$ | |
| 36: $\quad\quad\quad G \leftarrow G + [H(\mu'', \sigma'') - H(\mu''', \sigma''')]$ | |
| 37: $\quad \phi_a \leftarrow \phi_a - \alpha \nabla_{\phi_a} \mathbb{E}\left[G(\phi_a)\right]$ | |

sampling predictions. Delayed feedback and long-horizon requirements further challenge planning under the AIF framework. Additionally, many real-world tasks require rapid, frequent decisions and sustained performance in non-episodic, stochastic settings. To assess our approach, we employ a high-fidelity simulation environment validated to reflect realistic industrial control scenarios [55], which incorporates all the above challenges [11].

## 4.1 Application

We focus on simulating workstations in an automotive manufacturing system composed of parallel, identical machines (see Appendix for details). As energy efficiency becomes increasingly critical in manufacturing [56], RL offers a model-free alternative to traditional control, though it may struggle with rapid adaptations in non-stationary environments [57]. Governed by Poisson processes for arrivals, processing, failures, and repairs [55], the system evolves as a discrete-time Markov chain [58]. Control actions—switching machines on or off—aim to improve energy efficiency without compromising throughput. Due to stochastic delays, the system connects continuous-time dynamics to discrete-time decisions, making performance only observable over long horizons. Accordingly,

we employ a window-based preference metric [11] that evaluates KPIs over the past eight hours. The production rate is defined as $T = \frac{N(t) - N(t-t_s)}{t_s}$, where $N(t)$ is the number of parts produced, and the energy consumption rate as $E = \frac{C(t) - C(t-t_s)}{t_s}$, where $C(t)$ denotes total energy consumed, with $t - t_s \approx 8$ hrs. This window may span thousands of actions, where due to stochasticity and the integral nature of performance, immediate observations are noisy and uninformative. As a result, the AIF agents based on short-horizon EFE planning are not feasible in this setting. By operating directly on raw performance signals rather than handcrafted rewards, the approach enables scalability to domains where reward signals are sparse or expensive. The agent must handle delayed feedback and plan over extended horizons to move towards the preferred performance. This problem is continual with no terminal state, and decisions rely on both discrete and continuous observations.

## 4.2 Results

To validate the performance of our agent in the aforementioned environment, we adopted a rigorous evaluation scheme based on Algorithm 1. Unlike previous works that used batch interactions to improve training efficiency [12], our agent was trained in each epoch by interacting with a single environment instance, reflecting a more challenging setting. The trained agent's performance was then evaluated across several randomly initialized environments. From these, the best-performing instance was selected for a one-month simulation run to assess energy efficiency and production loss, in comparison to a baseline scenario where no control was applied and machines were continuously active. We also constructed a compositional preference score—analogous to a reward function—based on time-window KPIs for energy consumption and production, serving as an overall indicator of agent performance, which is part of the observation of the agent. To enforce further regularization in the latent space to match a normal distribution, we used a *Sigmoid* function in its non-saturated domain. Since we needed to encode the actor function, which is essentially a computational graph [50], we adopted a simple, non-parametric mapping $\Pi$ that concatenates the input with the first hidden and output values. Given its input-output structure and the fact that the model was continuously trained with that, this mapping effectively serves as an approximation of the actor's neural function (see Appendix for details on the agent and experimental setup).

We implemented the agent in the exact production system, using parameters verified to reflect realistic conditions, following the aforementioned scheme. Figure 1 presents the performance of the agent with an overshooting horizon of $H = 300$. During evaluations after each epoch (100 iterations), the agent improved the preference score of observations (Fig. 1a), which correlates with increased energy efficiency (Fig. 1b). Notably, the EEF of imagined trajectories used for policy updates decreased as the agent learned to control the system. This trend is observed in both the extrinsic and uncertainty components of the EFE. Since policy optimization relies heavily on learning a robust generative model—with the actor integrated within it—the agent gradually improved its predictive capacity and reduced reconstruction error across both discrete (Fig. 1d, preference) and continuous (Fig. 1e,f, machine and buffer states) elements of the observation space. While EFE and overall performance eventually stabilized, the generative model continued to improve, indicating that full reconstruction of future observations is not strictly required for effective control. The agent manages to improve the performance even when the overshooting horizon can be longer (e.g., $H = 1000$ steps; see Appendix). We then evaluated the trained agent over one month of simulated interaction (10 replications), applying gradient updates every $H$ steps during planning. Loffredo et al. (2023) [57] tested model-free RL agents across a reward parameter $\phi$, with DQN emerging as the top performer. Table 1 shows that our DAIF agent outstrips this baseline, raising energy efficiency per production unit by $10.21\% \pm 0.14\%$ while keeping throughput loss negligible.

## 5 Conclusion and Future Work

We introduced *Deep Active Inference Agents* (DAIF) that integrate a multi-step latent transition and an explicit, differentiable policy inside a single generative model. By overshooting the dynamics to a long horizon and back-propagating expected-free-energy gradients into the policy, the agent plans without an exhaustive tree search, scales naturally to continuous actions, and preserves the epistemic–exploitative balance that drives active inference. We evaluated DAIF on a high-fidelity industrial control problem whose feature complexity has rarely been tackled in previous works based on active inference. Empirically, DAIF closed the loop between model learning and control in highly

| Agent($\phi$) | Production Loss [%] | EN Saving [%] |
|---|---|---|
| DQN (0.93) | $4.82 \pm 0.34$ | $10.87 \pm 0.76$ |
| DQN (0.94 | $3.34 \pm 0.23$ | $9.92 \pm 0.69$ |
| **DAIF** | $\mathbf{2.59 \pm 0.16}$ | $\mathbf{12.49 \pm 0.04}$ |
| DQN (0.95) | $1.27 \pm 0.05$ | $7.00 \pm 0.07$ |
| DQN (0.96) | $1.27 \pm 0.09$ | $7.62 \pm 0.12$ |
| DQN (0.97) | $1.20 \pm 0.05$ | $7.72 \pm 0.10$ |
| DQN (0.98) | $0.54 \pm 0.04$ | $2.72 \pm 0.19$ |
| DQN (0.99) | $0.40 \pm 0.03$ | $2.46 \pm 0.01$ |

Table 1: Production loss versus energy-saving (EN) across reward parameters $\phi$ of DQN agents [57] and for the DAIF agent.

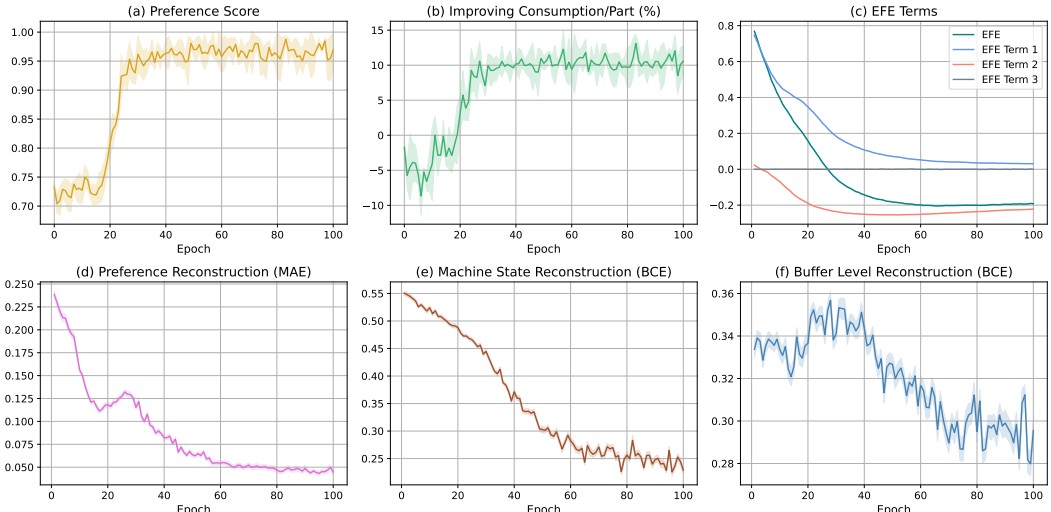

Figure 1: The performance of the agent with $H = 100$ on the real industrial system.

stochastic, delayed, long-horizon environment. With a single gradient update every $H$ steps, the trained agent planed, and achieved strong performance—surpassing model-free RL baseline—while its world model continued to refine predictive accuracy even after the policy stabilized.

**Limitations and future work:** While predicting an $H$-step transition removes the expensive *per-step* planning loop, the agent still has to gather *experience* after $H$ interactions and store it in the replay buffer for training, so its sample-efficiency can still be improved. To update the world model after each new environment interaction—obtained under different actor/moving parameters—we need an operator that aggregates the *sequence* of actor representations. Recurrent models are a natural choice for this, but their sequential unrolling adds latency and can hinder gradient flow. A lighter alternative is to treat the $H$ embeddings as an (almost) unordered set and use a set function [59]; when the temporal structure with simple positional embeddings (e.g. sinusoidal [60]) can be concatenated before the set pooling. This allows us to break the horizon into segments—down to a single step—and still backpropagate EFE gradients during planning through the aggregations the current policy representation. Finally, (neural) operator-learning techniques could enable resolution-invariant aggregation across function spaces [61, 62]. Additional extensions include replacing the VAE world model with diffusion- or flow-matching-based generators [28], adopting actor–critic optimization (as in Dreamer and related world-model agents [13, 6, 14]), and introducing regularization schemes to stabilize EFE gradient updates and reduce their variance. Rapid adaptation in non-stationary settings—where model-free agents often struggle—remains an especially promising direction.

Overall, this work bridges neuroscience-inspired active inference and contemporary world-model RL, demonstrating that a compact, end-to-end probabilistic agent can deliver efficient control in domains where hand-crafted rewards and step-wise planning are impractical.

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
