# OpenReview forum: "Deep Active Inference Agents for Delayed and Long-Horizon Environments"
_NeurIPS.cc/2025/Conference — Submitted to NeurIPS 2025_

### Official Review · Reviewer_oAWD · 2025-06-24

**Clarity:** 3
**Significance:** 3
**Originality:** 3
**Rating:** 4
**Confidence:** 3

**Summary:**

This paper scales active inference to long-horizon environments with delayed reward. It does this by leveraging key ideas from model-based reinforcement learning. While classical active inference approaches would struggle with long-horizon tasks due to computationally explosive planning, this approach finesses the problem by training a multi-step transition enabling a one-step horizon look-ahead. Further, classical tree searches are finessed by computing the gradients of expected free energy and minimizing those gradients using backpropagation using a planning neural network. In summary, the paper leverages key amortization ideas for scaling active inference. It then benchmarks the approach against deep Q-networks in a complex industrial task showing superior performance. Importantly, their baseline has been shown to be the best performing Model free baseline on this task in a previous paper.

**Questions:**

l352: 'the agent scales naturally to continuous actions,' : Can it scale to continuous observation or continuous state spaces as well? From l312 it seems that yes. This could be an important future direction since current active inference models in continuous space do not have the exploratory aspect of expected free energy in their dynamics.

'Perhaps one weak point is the benchmarking shown in Table 1. While the active inference agent performs well compared to the baseline from a recent paper, the benchmark could be more compelling if it were also shown against state-of-the-art model-based reinforcement learning algorithms like DREAMER or BBF, or compared to other state-of-the-art active inference algorithms, although it has been argued that such could not necessarily tackle the problem complexity explored here.' Could you explore additional benchmarks with canonical model-based RL algorithms or state-of-the-art active inference algorithms? Alternatively, could you say a bit more about why you chose a model-free algorithm as a benchmark for a model-based algorithm and why this is a suitable comparison?

typo l334: 'the EEF '

If I understand right, the two columns of table one are different reward modalities? It would be more compelling if reward was more emphasized here, so that the results become immediately compelling for a generic RL audience, and also, emphasise more that the DQN baseline is the strongest model-free baseline on this task.

typos: 'State epistemic uncertainty (Eq. 4a) ' should be 4b, 'Parameter epistemic uncertainty (Eq. 4a)' should be 4c.

typo l149: 'calculating this any possible'

typo 151: 'Fountas et al. (2020) [12] an auxiliary module'

typo l157: 'They also used the Kullback–Leibler
divergence between the planner’s policy and the habit provides as precision to modulate the latent
state [12 ]. They also used the Kullback–Leibler divergence between the planner’s policy and the habit159
provides as precision to modulate the latent state [ 12 ].'

typo l206: 'Sec. 2.1, which requires the tightly coupled modules illustrated in Fig. ??.'

This trained look-ahead transition - one of the main features of the paper - is very interesting. I wonder how this approach relates to another classic approach for dealing with long-horizon environments, namely, hierarchical generative models. Hierarchical genitive models have a degree of plausibility, and biological plausibility was one of the features of active inference that was emphasized in the introduction. Can you clarify on the respective strengths and weaknesses of your trained look-ahead transition approach vs. a hierarchical model approach? This would be particularly relevant for an active inference audience since hierarchical models are ubiquitous in this literature.

Because some key components of the architecture are not super clear from an active inference audience perspective, I just wanted to check that I correctly understand the concept of the trained long horizon transition: this is essentially using a neural net to predict the final state given a sequence of repeated actions over the whole horizon? The inclusion of the term amortization could be useful here.

One criticism of amortized approaches to planning is their inability to adapt to novelty. The paper keeps fast adaptation hooks on top of the amortized approach and shows the ability to interact with a complex environment. Furthermore, the limitations section addresses this. But perhaps this limitation/the trade-off between iterative and amortized approaches could be emphasized here (or elsewhere in the paper).

**Ethical Concerns:**

["NO or VERY MINOR ethics concerns only"]

**Final Justification:**

The authors have addressed all my questions. Based on my original assessment, I am pleased recommend the paper for its novelty, and because I believe the ideas herein would be useful for the community, for scaling RL and active inference. One important point for which I would like to know the details (but these will be released in the camera-ready version as noted by the authors) is additional benchmarking on new environments, which will provide more clarity on the robustness of the method in several settings. The main unresolved experimental point is the fact that the work does not at this stage contain all the bells and whistles needed for a favourable head-to-head comparison with state-of-the-art model-based RL approaches, as noted by the authors. For this reason, I rate the paper at 4.

**Limitations:**

yes

**Quality:**

3

**Strengths And Weaknesses:**

One strong point of this paper is the application of active inference to problems that are more complex than many previously tackled in the literature. By operating directly on raw performance signals rather than handcrafted rewards, the approach enables scalability to domains where reward signals are sparse or expensive.

The paper is compelling in its use of amortized approaches for scaling active inference methods while retaining the epistemic drive characteristic of active inference agents. I expect that these ideas could be very useful moving forward.

Perhaps one weak point is the benchmarking shown in Table 1. While the active inference agent performs well compared to the baseline from a recent paper, the benchmark could be more compelling if it were also shown against state-of-the-art model-based reinforcement learning algorithms like DREAMER or BBF, or compared to other state-of-the-art active inference algorithms, although it has been argued that such could not necessarily tackle the problem complexity explored here. I've added some suggestions in the next section.

Results indicate that ' full reconstruction of future observations is not strictly required for effective control'. This is an interesting observation.

Clarity: the paper is well written and reasonably clear. From an active inference reader's perspective though, there are many RL terms/background literature that are not intimately familiar/explained in familiar terms. This is alright though, but I have listed a few questions in the next section.

One criticism of the paper is that one of the core features, namely the trained learned transition, is lifted from a previous paper but not original to this paper.

The description of limitations of the paper seems comprehensive.

---

> ### Author Rebuttal · Authors · 2025-07-31
>
> We thank you for your detailed and supportive review. We also appreciate your recognition of our contribution. We agree that the ideas developed in this work can be valuable moving forward, potentially enabling powerful and practical agents based on active inference. We respond to your points below:
>
> **Terminology:** Given that our work lies at the intersection of Active Inference, reinforcement learning, and deep generative modeling, we acknowledge that some degree of terminology overload may have occurred. To address this and enhance clarity, we will include a glossary table of key terms and notations in the supplementary materials of the camera-ready version. We hope this will make the paper more accessible to readers across different disciplines.
>
> **Learned transition:** We would like to highlight that a key contribution of our work—compared to prior literature—is the integrating of the policy (network) directly into the transition model, enabling multi-step overshooting. This design supports efficient computation of gradients of the EFE, which is central to our policy training and planning.
>
> The trained transition is implemented as a neural network that predicts the final state given the initial state and a policy embedding. Specifically, $P_{\theta_s}\left(s_{t+H} \mid \widetilde{s_t}, \hat{\pi} \right)$ is amortized inference, producing a distribution over $s_{t+H}$ given the sampled latent state at time~$t$, while the actor—denoted by embedding $\hat{\pi}(\phi_a)$—is assumed fixed throughout the horizon. By embedding the policy directly into the transition (rather than using a sequence of actions), we enable policy gradients based on optimization of the EFE.
>
> **Typos**: We thank you for kindly pointing them out and sincerely apologize for the oversights. We have carefully proofread the manuscript and will address all inconsistencies—including those you helpfully noted—in the camera-ready version.
>
> *Architecture Figure:* The figure referenced in line 206 is included as Figure 2 in the supplementary materials.
>
> **Scalability to continuous spaces:** Yes. Our framework scales naturally to continuous observation and state spaces in addition to continuous actions, whereas some of the prior literature is restricted to discrete settings. In our experiments the observation vector already contains a mix of discrete and continuous elements (please see B.2.1 of the supplement), and the latent state is modeled with Gaussian distributions. All three terms of the EFE—including the epistemic (exploratory) components—are retained and differentiable in this setting. We agree with you that such scalability is crucial, since many real world problems cannot be cast in a purely discrete form. Likewise further analyzing how exploration behaves is indeed an important avenue for future work.
>
> **Benchmark against model-free baselines:** We employed the task carefully benchmarked in [3] against several model-free RL agents, including DQN, TRPO, and PPO. Among them, DQN was shown to be the best-performing agent. In parallel, [3] also derived an approximate mathematical solution for a specific scenario using linear programming (LP), made possible by explicitly modeling the system dynamics and parameters as a discrete-time Markov decision process. For a constrained control scenario—specifically, maintaining throughput loss below 1.15%—the LP-based optimal solution yielded an energy saving very close to the DQN agent’s performance. This highlights that, despite being model-free, DQN can serve as a strong baseline for control tasks on this environment.
>
> The simulator (environment) was carefully developed and validated to reflect the behavior of a real industrial system. Importantly, it represents a generic industrial automation problem that has been shown to scale to different configurations and stages [4].
>
> It is also worth noting that, in our work, we reformulated the objective function via a preference mapping (please see Supplement A.3). We employed a window-based approach (covering eight simulation hours, ~3000 actions) to incorporate raw performance metrics directly into the agent’s preferences. This formulation increases the temporal abstraction and complexity of the problem relative to the original reward function in [3], challenging any RL baselines. This setup emphasizes on delayed and long-horizon conditions, specifically to demonstrate the effectiveness of the Deep AIF agent in such challenging settings.
>
> **Additional benchmark and comparisons:** The work presented here is a proof of concept for applying policy gradients based on EFE using an integrated policy within a generative model—bridging Active Inference and RL through generative modeling. We acknowledge that state-of-the-art model-based RL agents, such as Dreamer [1–2], employ several signal-shaping techniques (e.g., actor–critic architectures, normalization layers, and loss-balancing strategies) to achieve robust training. In contrast, our current Deep AIF agent is minimal and does not yet incorporate these enhancements—though it can be extended to do so. As such, a strict one-to-one comparison of superiority would not be entirely fair, as it compares a generic proof-of-concept framework to highly enhanced agent implementations. In future Deep AIF instances, one could integrate policy training strategies from RL, improved EFE estimation methods from Active Inference, and more advanced generative modeling techniques from deep learning (e.g., diffusion models or flow matching).
>
> That said, we agree that broader benchmarking against canonical model based (and model free agents) would strengthen the paper. Following you valuable suggestion and  leveraging the agent’s scalability, we are conducting additional experiments in other benchmarks to be included in (the supplementary materials) of the camera ready version.
>
> **Reward/Performance Modalities:** Thank you for pointing this out. We agree that reporting rewards can be more compelling for an RL audience. However, in the context of our application, this is not straightforward due to the complexity of how rewards and preferences are defined across agents. As shown in Table 1, each agent is trained with a different reward or preference function (see Supplement A.3), and these functions have different scales. For model-free baselines from [3], the reward depends on a parameter $\phi$ that balances energy savings against throughput with preference function used for training DAIF being conceptually different.
>
> Also, to accurately evaluate agent performance, we need a careful and consistent evaluation protocol (see Supplement C.2) that involves deploying each trained agent in a separate long-term simulation of 30 days. It provides a shared evaluation metric that is independent of the training objective.
>
> **Comparisons with HGMs**: We thank the reviewer for insightfully raising this point. We agree that there are similarities between our approach and hierarchical generative models (HGMs). As shown in Figure 2 of the supplementary materials, the overall architecture of Deep AIF can be viewed as having two levels: the lower level (policy) handles step-by-step navigation, connecting to the higher level corresponds to the learned generative model with longer temporal transitions—similar in spirit to classical HGMs such as [5], and aligning with notions of biological plausibility. However, despite this conceptual analogy, distinctions exist. In our framework, long-horizon inference is amortized through a single-step transition model that directly predicts future states over a fixed horizon. This eliminates the need for recursive inference or message-passing procedures typically required in HGMs, leading to significant advantages in computational efficiency, training stability, and architectural simplicity.
>
> Leveraging the expressiveness of neural networks, Deep AIF can approximate a wide range of computational behaviors in an end-to-end differentiable manner. That said, this approach trades off some of the explicit interpretability and structured multi-scale reasoning afforded by HGMs.
>
> **Fast adaptations:** We thank you for valuably pointing this out. We agree on the importance of this aspect and will emphasize it in the limitations section of the camera-ready version.
>
> --
>
> [1] Hafner, D., Lillicrap, T., Ba, J., & Norouzi, M. (2019). Dream to control: Learning behaviors by latent imagination. arXiv preprint arXiv:1912.01603.
>
> [2] Hafner, D., Pasukonis, J., Ba, J., & Lillicrap, T. (2025). Mastering diverse control tasks through world models. Nature, 1-7.
>
> [3] Loffredo, A., May, M. C., Schäfer, L., Matta, A., & Lanza, G. (2023). Reinforcement learning for energy-efficient control of parallel and identical machines. CIRP Journal of Manufacturing Science and Technology, 44, 91-103.
>
> [4] Loffredo, A., May, M. C., Matta, A., & Lanza, G. (2024). Reinforcement learning for sustainability enhancement of production lines. Journal of Intelligent Manufacturing, 35(8), 3775-3791.
>
> [5] Van de Maele, T., Dhoedt, B., Verbelen, T., & Pezzulo, G. (2024). A hierarchical active inference model of spatial alternation tasks and the hippocampal-prefrontal circuit. Nature Communications, 15(1), 9892.

---

> > ### Comment · Reviewer_oAWD · 2025-08-06
> >
> > Thank you for your detailed answer. I believe all of my points have been addressed, and I am pleased to recommend the paper for acceptance. I look forward to seeing the additional experiments the authors are conducting in the camera-ready version.

---

> ### Author Response · Authors · 2025-08-09
>
> Dear Reviewer,
>
> Thank you for your meaningful support and constructive comments. We are committed to adding the additional benchmarks and experiments in the camera-ready version.
>
> We would like to appreciate your thoughtful review and the time you invested in this work; your feedback is valuable in advancing the field.

---

### Official Review · Reviewer_mpAY · 2025-07-01

**Clarity:** 2
**Significance:** 2
**Originality:** 3
**Rating:** 3
**Confidence:** 4

**Summary:**

The authors introduce Deep Active Inference Agents (DAIF), an attempt to scale active-inference (AIF) control to long-horizon settings. The central idea is to embed a differentiable policy network inside the agent’s generative model and to propagate it through an “overshoot” transition that jumps $H$ steps ahead in latent space. Expected Free Energy (EFE) computed over this horizon not only trains the world-model (via variational free energy) but also supplies direct gradients to the policy, sidestepping the tree-search used in earlier discrete AIF work. The authors demonstrate the method on a single manufacturing-line simulator in which machinery must be switched on and off to minimise energy per produced part. Against a DQN baseline the DAIF agent achieves an energy saving whilst maintaining throughput, and the accompanying repository (anonymised for review) promises reproducibility.

**Questions:**

Have you evaluated DAIF against modern world-model agents that also tackle long horizons?
How sensitive is performance to
(i) the overshoot horizon,
(ii) embedding the actor inside the generative model, and
(iii) the choice of preference mapping?
Can you articulate, empirically or theoretically, why EFE provides a better training signal than conventional value-function gradients in comparable world-model agents?
Could you provide the figure mentioned in the broken reference? “Figure ??”.
What hyperparameter tuning was performed on the DQN baselines?

**Ethical Concerns:**

["NO or VERY MINOR ethics concerns only"]

**Final Justification:**

I thank the authors for their careful response to my original review. I believe however that the main criticism remains in that for Neurips, a proof of concept on a very limited domain is not sufficient to show that this is a useful, generalisable result. While it may indeed be, this needs to be shown by comparing with a range of SOTA algorithms on a range of problems, not necessarily to show superiority, but to show that it is indeed a useful framework. As it currently stands, I don't believe that this has been shown strongly enough for Neurips main track. The statement that comparisons have been made to DAIMC is useful, but the lack of detail of these comparisons in the paper makes it hard to judge precisely what has been done. Given the responses I have upgraded my rating to a 3.

**Limitations:**

Yes

**Paper Formatting Concerns:**

N/A beyond those noted above.

**Quality:**

2

**Strengths And Weaknesses:**

The theoretical component is decent: the paper carefully derives how EFE gradients can stand in for explicit planning, and demonstrates that the policy-in-model trick is mathematically consistent with active inference principles. Code is to be released, aiding reproducibility.

Experimentally, the authors show a statistically significant improvement over the best–tuned DQN baseline: Table~1 reports an energy saving of 12.49±0.04% for DAIF versus 10.87±0.76% for the strongest DQN configuration.

Although the absolute gain (~1.6 percentage points) is modest, the result is clearly more than noise.  The evaluation, however, remains narrow - there is only one task and no comparisons against state-of-the-art world-model RL baselines.

The manuscript is logically organised and figures generally support the text. That said, awkward phrasing, broken cross-references (e.g., Line 206 “Figure ??”, Lines 125, 129. and 131 all points to Eq. 4a”), and contradictory horizon values (Lines 331-332 H = 300 in the prose, and H=100 in Figure 1) impede readability. Much of the terminology presumes familiarity with active inference, so non-specialist NeurIPS readers may struggle. A tighter edit, plus an explicit comparison to standard RL language, would help.

Tree-search world–model agents such as AlphaZero, MuZero and EfficientZero have delivered state‐of‐the‐art performance on extremely long‐horizon problems by explicitly expanding a search tree at every decision step. That success shows that when computation is plentiful, planning can yield impressive returns.

The contribution of the present paper lies in a different trade–off: it removes tree search altogether and instead back‐propagates EFE through a multi‐step latent transition. Doing so keeps the per–decision cost constant regardless of the horizon’s branching factor and, crucially, allows the policy to output continuous control variables (torques, valve openings, steering angles) without first discretising them for enumeration.

If the authors can demonstrate that this fixed, differentiable planning scheme matches or approaches the decision quality of tree–search agents on standard benchmarks while using fewer computational resources, the work would have substantial practical impact. At present, however, no head‐to‐head comparison with modern tree–search baselines is provided, so the broader significance remains speculative.

Within the AIF community, embedding the actor in the generative model and overshooting many steps appears to be novel.  In the wider model‐based RL landscape the technique resembles existing world‐model agents (is correctly referenced), with the principal novelty being the use of EFE rather than value gradients.  The paper does not yet show why this switch confers a clear advantage beyond the reported industrial case.

In short, the work offers a well‐motivated and now demonstrably significant improvement over a baseline, but its empirical scope is still too limited, and the presentation needs refinement, for main‐track NeurIPS acceptance.

---

> ### Author Rebuttal · Authors · 2025-07-31
>
> Thank you for your insightful review. We respond to your points below:
>
> **DAIF against model-free baselines:** The application we studied has previously been benchmarked using several model-free RL agents, including PPO, TRPO, and DQN, with the latter emerging as the best-performing baseline [1]. We therefore presented comparison in Table 1 on DQN, as the reward function used in that study [1] includes a hyperparameter that balances energy savings against production throughput. It is important to note that energy savings should be interpreted in the context of production loss—i.e., higher savings are meaningful if throughput is not significantly compromised. For example, DAIF achieves 12.49% energy savings with a 2.59% production loss, compared to the closest of DQN’s with a 3.34% loss and 9.92% saving.
>
> Additionally, DQN and the other baselines were tuned using a grid search over the reward function’s hyperparameters [1]. It worth noting, [1] also derived an approximate mathematical solution for a specific scenario using linear programming (LP), made possible by explicitly modeling the system dynamics and parameters as a discrete-time Markov decision process, For a constrained control scenario—specifically, maintaining throughput loss below 1.15%—the LP-based optimal solution yielded an energy saving very close to the DQN agent’s performance. This highlights that, despite being model-free, DQN can serve as a strong baseline for control tasks on this environment.
>
> In contrast, both the policy architecture and training hyperparameters for the Deep AIF agent were used out-of-the-box and were not tuned, further highlighting the robustness of our approach despite minimal optimization.
>
> **Writing oversights:** We thank you for noting these issues and sincerely apologize for the oversights. We have carefully proofread the manuscript and will address all inconsistencies in the camera-ready version. The figure referenced in line 206 was included as Figure 2 in the supplementary materials. Lines 129 and 131 should refer to Eq. 4b and Eq. 4c, respectively. Additionally, the value of H in Figure 1 should be 300, not 100.
>
> *Architecture Figure:* The figure referenced in line 206 is included as Figure 2 in the supplementary materials.
>
> **Terminology:** since our work lies at the intersection of Active Inference, RL, and deep generative modeling, some degree of notation and terminology overload may have occurred. In line with your helpful suggestion to ensure consistency with the RL language, we will include a glossary table of key terms and notations in the supplementary materials to further improve clarity and readability.
>
> **Comparison with Tree-search world-models:** We agree with you that state-of-the-art tree-search world-model agents have shown strong performance. However, such methods may still struggle in our problem setting. For instance, we reimplemented and adapted DAIMC [4]—a tree-search world-model agent empowered with active inference, known for good performance. We adapted DAIMC for the problem and experimented with several hyperparameter settings; However, it was unable to exert any meaningful control over the system—even after substantial runtime—performing comparably to a random policy. DAIMC relies on computationally expensive MCTS with only one- or few-step lookahead, which is insufficient in our long-horizon, stochastic setting where meaningful changes emerge only after tens to hundreds of actions. Even with expanded search depth, the planner remained ineffective, largely reacting to stochastic noise rather than underlying dynamics. In contrast, our agent overcomes these limitations by (i) using a multi-step latent transition to capture long-horizon dynamics in a single forward pass, and (ii) embedding a differentiable policy within the generative model, enabling efficient gradient-based planning with minimal computation cost. This design allows our agent to operate effectively in delayed and long-horizon scenarios without costly tree search.
>
> It is also possible to develop hybrid methods that combine our current overshooting generative model with tree search, effectively increasing the planning depth while still producing differentiable policy gradient signals.
>
> **Empirical scope and comparison with world-model RL baselines:** The work presented serves as a proof of concept for applying policy gradients based on the EFE using an integrated policy within a world model—effectively bridging Active Inference and RL through generative modeling. We acknowledge that state-of-the-art world-model RL baselines, such as Dreamer [2–3], employ several signal-shaping techniques (e.g., actor–critic architectures, normalization layers, and loss-balancing strategies) to achieve robust training. In contrast, our current Deep AIF agent is minimal and does not yet incorporate these enhancements—though it can be extended to do so. As such, a strict one-to-one comparison of superiority would not be entirely fair, as it compares a generic proof-of-concept framework to highly enhanced agent implementations.
> Our empirical goal was to demonstrate that the proposed concept is effective in long-horizon and delayed scenarios. Therefore, we showed effective control in an environment that was carefully developed and validated to reflect the behavior of a real industrial system [1]. Importantly, this simulator represents a generic industrial automation problem, which has been shown to scale across different configurations and production stages. In future Deep AIF instances, one could integrate policy training strategies from RL, improved EFE estimation methods from Active Inference, and more advanced generative modeling techniques from deep learning (e.g., diffusion models or flow matching)
>
> That said, we agree that broader benchmarking against world-model and RL baselines would strengthen the paper. Following you valuable suggestion and leveraging the agent’s scalability, we are conducting additional experiments in other benchmarks to be included in (the supplementary materials) of the camera ready version.
>
> **Novelty:** We appreciate you recognizing the theoretical component and novelty of the work. In addition to contribution to AIF, we think that our approach still differs from commonly used methods in the model-based RL landscape—such as the well-known Dreamer [2–3]—in a fundamental way: rather than performing multiple sequential rollouts in latent (dreamed) space, we directly embed the policy into the generative model and use multi-step latent overshooting to predict the future in a single look-ahead, with carefully-derived EFE accounting both for extrinsic and epistemic drive.
>
> **Sensitivity of the performance:**
>
> - We studied the effect of the overshooting horizon in Section C.3 of the supplementary materials, which shows that the framework remains effective across a range of planning depths.
> - For actor embedding, we currently use a simplified and approximate approach by directly incorporating the neural network parameters. While more sophisticated embeddings—such as those based on GNNs or neural implicit representations—could be used, we intentionally avoided them in this proof-of-concept to keep the architecture and training tractable.
> - We confirm that the agent’s performance is sensitive to the preference function, which differs in logic from standard reward functions and includes its own set of hyperparameters. Since the agent is driven toward of higher preferences, the form of the preference function and the choice of goal setpoints influence performance. In Section A.3 of the supplementary materials, we included two different forms of the preference function, both of which yielded very similar control performance in the case reported in Section 4.2.
>
> **EFE training signal:** The gradient-bases optimization of the EFE within the proposed world-model architecture offers the following promises:
> - Derived from a Bayesian-grounded framework (heuristic-free)
> - Natively scalable to continuous action spaces
> - A unified explore–exploit gradient
> - Effective encoding of long-horizon dynamics
> - Capturing distinct forms of epistemic uncertainty, providing an intrinsic drive for model refinement
> - A degree of interpretability
> - Reliance on observations (cheap, raw data), instead of often expensive, engineered reward signals
> - Inherent adaptability in non-stationary settings via AIF
> - Minimal computational cost during both inference (model-free) and planning (EFE calculated in a single H-step forward pass)
>
> --
>
> [1] Loffredo, A., May, M. C., Schäfer, L., Matta, A., & Lanza, G. (2023). Reinforcement learning for energy-efficient control of parallel and identical machines. CIRP Journal of Manufacturing Science and Technology, 44, 91-103.
>
> [2] Hafner, D., Lillicrap, T., Ba, J., & Norouzi, M. (2019). Dream to control: Learning behaviors by latent imagination. arXiv preprint arXiv:1912.01603.
>
> [3] Hafner, D., Pasukonis, J., Ba, J., & Lillicrap, T. (2025). Mastering diverse control tasks through world models. Nature, 1-7.
>
> [4] Fountas, Z., Sajid, N., Mediano, P., & Friston, K. (2020). Deep active inference agents using Monte-Carlo methods. Advances in neural information processing systems, 33, 11662-11675.

---

> > ### Comment · Reviewer_mpAY · 2025-08-05
> > **Final justification note**
> >
> > I thank the authors for their careful response to my original review. I believe however that the main criticism remains in that for Neurips, a proof of concept on a very limited domain is not sufficient to show that this is a useful, generalisable result. While it may indeed be, this needs to be shown by comparing with a range of SOTA algorithms on a range of problems, not necessarily to show superiority, but to show that it is indeed a useful framework. As it currently stands, I don't believe that this has been shown strongly enough for Neurips main track. The statement that comparisons have been made to DAIMC is useful, but the lack of detail of these comparisons in the paper makes it hard to judge precisely what has been done. Given the responses I have upgraded my rating to a 3.

---

> ### Author Response · Authors · 2025-08-09
>
> Dear Reviewer,
>
> Thank you for your thoughtful points.
>
> **Comparison with DAIMC**:
>
> We conducted extensive experiments of DAIMC, including many adaptations to the manufacturing problem. Notably:
>
> One-step selection (no planner): Using DAIMC’s original one-step selection, we obtained 0.664 ± 0.021, comparable to a random agent under our current preference settings.
>
> MCTS adaptation: We integrated MCTS into the core DAIMC and varied tree-search hyperparameters and preference forms. We started with a simulation depth of 5 (appropriate for our long-horizon, high-frequency decisions), tried different threshold levels, and toggled habit features. We also retrained the task reward to the original, problem-specific formulation (differential production and energy saving), which increases short-horizon variability—favorable to MCTS.
>
> Outcome: Despite these adaptations, the planner exhibited fluctuating behavior and failed to maintain any control, while incurring heavy EFE computations at every step. Representative test metrics:
>
> Throughput: 1.72 parts/min
> Energy per part: 1453.09 kJ/part
>
> These are far below the proposed DeepAIF controller, which preserves high throughput with lower energy:
>
> Throughput: DAIF = 2.9296 parts/min vs. ALL-ON = 2.98 parts/min
>
> Energy per part: DAIF = 1425.16 kJ/part vs. ALL-ON = 1595.00 kJ/part
>
> Further scaling: We extended search depth to 50+ and trained for 400 epochs (>20 days), tested multiple habit formulations, and even replaced habit with Q-learning. Nevertheless, MCTS still failed to yield stable control—rendering it intractable for real industrial settings with hundreds of machines and online decision demands (even with only 6 machines, the action space is ~3000 actions in each shift).
>
> Conclusion: In our scenario, MCTS primarily tracks system stochasticity rather than enabling effective control. Our DAIF setup further stresses delayed and long-horizon with an 8-hour preference window.
>
> We intentionally omitted these details in the paper to avoid over-emphasizing negative results for DAIMC, which we still view as a valuable contribution (we adopted elements of its EFE computation).
>
> **Usefulness**:
>
> We respectfully offer a our view of usefulness. DAIMC, previously accepted at NeurIPS main track, evaluated two visual benchmark environments (naturally aligned with biologically inspired agents) and compared against standard model-free agents. By contrast, our environment has additional peculiarities and complexities, crucially, involves a real-world control application. Although DAIMC did not solve this task, we found it useful science, and we incorporated its key EFE ideas. More generally, we note that the same “not useful” razor has at times been raised—even at initial submission—against seminal work, including Physics-Informed Neural Networks [1], which later had substantial impact. We believe usefulness lies in scientific contribution—the why and how—not solely in immediate SOTA dominance.
>
> Importantly, DeepAIF is implemented in a verified, realistic simulator, and the results demonstrate practical applicability, enabling immediate deployment without further adjustments.
>
> We are committed to providing robust benchmarks of DeepAIF against both model-based and model-free SOTA methods in our camera-ready version.
>
> [1] Raissi, M., Perdikaris, P., & Karniadakis, G. E. (2019). Physics-informed neural networks: A deep learning framework for solving forward and inverse problems involving nonlinear partial differential equations. Journal of Computational Physics, 378, 686–707.
>
> **Finally**, we appreciate your thoughtful review and the time invested in reviewing this work; your feedback helps advance the field.

---

### Official Review · Reviewer_sddb · 2025-07-03

**Clarity:** 2
**Significance:** 3
**Originality:** 3
**Rating:** 4
**Confidence:** 3

**Summary:**

In this paper, the authors present an integrated architecture for world modeling, planning and learning based on the Active Inference framework from neuroscience. Their main contribution relies in introducing long-horizon world modeling to handle delayed effects and long-term credit assignment. They empirically evaluate their method on an energy optimization problem for manufacturing and show the effectiveness of their method with respect to a DQN agent.

**Questions:**

- Where do the preferences come in? It makes reference to Eq. 4a but I don’t understand where $\Psi$ comes in that equation
- I assume that $\hat{\pi}$ is the embedding of the policy function and that $\mathcal{Q}_{\phi_a}$ is the space of all policy functions? But why is that set subscripted  by $\phi_a$? Aren’t these the NN parameters that represent a function $Q \in \mathcal{Q}$?
- In line 232, what is $G_\theta(\tilde{o}, \phi_a)$? I thought $G$ was supposed to be the EFE? Why is this a function now of the observation instead of a functional of the policy *and* time step?

**Ethical Concerns:**

["NO or VERY MINOR ethics concerns only"]

**Final Justification:**

I will keep my current score of 4. I believe the authors addressed my concerns and I believe that overall is an interesting framework that gives an alternative framing of decision-making. Though it's still a proof of concepts there are interesting ideas to be published.

**Limitations:**

yes

**Paper Formatting Concerns:**

No major formatting concerns

**Quality:**

2

**Strengths And Weaknesses:**

Strengths
- Their approaches leverages the active inference framework for decision-making and further incorporates long-horizon world modelling and planning.
- They introduce a novel approach to modeling a multi-step world model in a VAE-framework without recursive simulation by conditioning on an embedding of the policy function.

Weaknesses
- The paper’s writing is sometimes hard to follow. Notational inconsistencies and/or overloading hinder the reading of the paper.
- In their empirical evaluation they do not compare with previous AIF implementations (e.g., Fountas et al, 2022) that could allow to understand how the long-horizon modeling introduce in this work improves over previous more myopic implementations
- It would be interesting to compare to MBRL methods? How does this compare with this AIF approach?
- Some citations missing in works that do RL with world models for long-horizon modeling: * Bagaria et al. Skill Discovery for Exploration and Planning using Deep Skill Graphs (ICML 2021);  *Freed et al. Learning temporally AbstractWorld models without online experimentation (ICML 2023); * Rodriguez-Sanchez et al. Learning Abstract World Models for Value-preserving Planning with Options (RLC 2024).

---

> ### Author Rebuttal · Authors · 2025-07-31
>
> Thank you for your constructive review. We respond to your points below:
>
> **Writing and notation:** We appreciate your note on this. We have carefully proofread the manuscript and will address all inconsistencies in the camera-ready version. Additionally, we will include a glossary table with definitions of all notations in the supplementary materials to further improve readability.
>
> **Comparison with previous AIF implementations:** We have carefully considered previous AIF implementations, notably DAIMC [1], which demonstrated strong performance against model-free RL in visual control tasks. We adapted DAIMC for the problem and experimented with several hyperparameter settings; However, it was unable to exert any meaningful control over the system—even after substantial runtime—performing comparably to a random policy. It relied on computationally expensive MCTS with only one- or few-step lookahead in its generative model. In our problem, meaningful changes emerge only after tens to hundreds of actions due to delayed and stochastic feedback. As a result, DAIMC’s myopic prediction and planning mechanism were ineffective in this setting. In contrast, our proposed agent addresses both of these limitations by (i) using a multi-step latent transition that captures long-horizon dynamics in a single look-ahead, and (ii) embedding a differentiable policy directly in the generative model, enabling gradient-based planning instead of expensive search. This allows our agent to operate effectively in the delayed, long-horizon setting.
>
> **Comparison with MBRL methods:** The work presented here is a proof of concept for applying policy gradients based on EFE using an integrated policy within a world/generative model—bridging Active Inference and RL through generative modeling. We acknowledge that state-of-the-art MBRLs agents, such as Dreamer [2–3], employ several signal-shaping techniques (e.g., actor–critic architectures, normalization layers, and loss-balancing strategies) to achieve robust training. In contrast, our current Deep AIF agent is minimal and does not yet incorporate these enhancements—though it can be extended to do so. As such, a strict one-to-one comparison of superiority would not be entirely fair, as it compares a generic proof-of-concept framework to highly enhanced agent implementations.
> In future Deep AIF instances, one could integrate policy training strategies from RL, improved EFE estimation methods from Active Inference, and more advanced generative modeling techniques from deep learning (e.g., diffusion models or flow matching).
>
> That said, we agree that broader benchmarking against MBRL baselines would strengthen the paper. Following your valuable suggestion and leveraging the agent’s scalability, we are conducting additional experiments in other benchmarks to be included in (the supplementary materials) of the camera ready version.
>
> **Comparison between MBRL and AIF approaches:** We argue that while there are conceptual similarities—such as learning a world model and using it for action selection—their mathematical foundations differ. AIF is derived from a variational principle applied to a generative model integrated with a policy, offering several inherent theoretical promises, as discussed in Sections 1 and 2. One notable strength lies in real-time responsiveness: AIF naturally incorporates preferences (e.g., setpoints or goal states), making it more aligned with control paradigms, particularly in non-stationary or dynamically changing environments.
>
> **Missing citations:** We sincerely thank you for pointing out the missing citations [4-6] related to long-horizon world-model RL. We will ensure they are properly cited in the camera-ready version.
>
> **Preference:** Under the AIF framework, agents require a prior preference over predictions to guide behavior, which is formalized as the extrinsic value in Eq. 4a (i.e.,$-\mathbb{E}[\log P(o_\tau \mid \pi)]$). In fact, during computation of the EFE, it is therefore necessary to evaluate how closely the predicted distribution aligns with the agent's preferred observations. To achieve this, we define an analytical mapping that transforms the prediction distribution into a continuous preference spectrum, $\Psi : P_{\theta}(o_{\tau}) \rightarrow [0, 1]$.
> . This transformation is applied only during the EFE computation and is thus not explicitly visible in the formalism itself.
>
> **Policy notation:** We thank you for pointing this out and apologize for the confusion—the calligraphic symbol $\mathcal{Q}$ was a typo and should be $Q$. We confirm that $Q_{\phi_a}$ denotes the policy function (or actor), parameterized by $\phi_a$, and that the mapping $\Pi: Q_{\phi_a} \rightarrow \hat{\pi}$ produces the encoded representation $\hat{\pi}(\phi_a)$ used within the generative model.
>
> **EFE computation:** When computing the EFE—denoted by $G$—the generative model, parameterized by $\theta$, produces predictions starting from an observation $\widetilde{o}$, generating a distribution over a horizon of $H$ steps, conditioned on the policy parameters $\phi_a$. Accordingly, the EFE is expressed as $G_\theta\left(\widetilde{o}, \phi_a\right)$, making it a function of both the initial observation and the policy parameters. This formulation enables differentiation with respect to $\phi_a$, facilitating direct optimization of the policy via gradient descent. Importantly, treating $G$ as a function of observations also allows for batch-wise training of the policy, where gradients can be computed from a set of observations or scenarios.
>
> --
>
> [1] Fountas, Z., Sajid, N., Mediano, P., & Friston, K. (2020). Deep active inference agents using Monte-Carlo methods. Advances in neural information processing systems, 33, 11662-11675.
>
> [2] Hafner, D., Lillicrap, T., Ba, J., & Norouzi, M. (2019). Dream to control: Learning behaviors by latent imagination. arXiv preprint arXiv:1912.01603.
>
> [3] Hafner, D., Pasukonis, J., Ba, J., & Lillicrap, T. (2025). Mastering diverse control tasks through world models. Nature, 1-7.
>
> [4] Bagaria, A., Senthil, J. K., & Konidaris, G. (2021, July). Skill discovery for exploration and planning using deep skill graphs. In International conference on machine learning (pp. 521-531). PMLR.
>
> [5] Freed, B., Venkatraman, S., Sartoretti, G. A., Schneider, J., & Choset, H. (2023, July). Learning temporally abstractworld models without online experimentation. In International Conference on Machine Learning (pp. 10338-10356). PMLR.
>
> [6] Rodriguez-Sanchez, R., & Konidaris, G. (2024). Learning Abstract World Model for Value-preserving Planning with Options. RLC 2024.

---

> > ### Comment · Reviewer_sddb · 2025-08-06
> >
> > Thank you for addressing my questions. I appreciate it!
> > Given the responses I'll keep my current score. I believe that the framework is an interesting paradigm.
> > Also, I would encourage the authors to add the comparison to MBRL and extending the discussion on the relationship between this framework and MBRL.

---

> > > ### Author Response · Authors · 2025-08-09
> > >
> > > Dear Reviewer,
> > >
> > > Thank you for your recognition of our work and for the constructive comments. We are committed to adding additional benchmarks and experiments—particularly with MBRL—and to extending the discussion to further clarify the connection between our framework and MBRL, as you thoughtfully suggested, in the camera-ready version.
> > >
> > > We appreciate your thoughtful review and the time you invested; your feedback helps advancing the field.

---

### Note · Authors · 2025-08-16

Dear Area Chair,

We offer the following remark, without repeating earlier discussion: in line with the reviewers’ recognition and comments, the proposed concept—bridging world/generative models, Active Inference, and RL—has potential utility across diverse tasks and settings, connecting to state-of-the-art techniques. The work also contributes to scaling Active Inference—grounded in neuroscience and supported by a clear mathematical formulation—into practical agent architectures consistent with current ML practice.

The experimental design—and the complex, validated real-world environment—directly matches the paper’s stated focus on delayed, long-horizon settings. With its particularities, it is a challenging and representative testbed for this class of problems and was used to validate the paper’s concepts. Beyond this initial experimental scope, the AIF world-model policy-gradient idea opens multiple avenues for agent instances and analysis (e.g., policy encoding, signal shaping/regularization, generative-model technique, AIF-specific choices). Relative to highly tuned SOTA models on diverse benchmarks, these become opportunities and broaden the scope. We will include results across additional benchmarks/settings in the camera-ready version with initial agent instances, and we expect this to remain an active research direction

**Finally**, we sincerely appreciate your efforts, and those of the senior and program chairs. We strongly believe these collective efforts advance the field.

---

### Decision · Program_Chairs · 2025-09-17

**Decision:**

Reject

**Comment:**

(a) The paper proposes deep active inference agents (DAIF), bridging active inference and RL. The proposed method first learns a multi-step future in one-shot, embeds a policy inside the generative models with back propagation (via the “expected free energy” or EFE term) and trains the model/policy in an iterative fashion. Empirical experiments are show on one environment consisting of an industrial manufacturing simulator with delayed feedback and long horizons. DAIF is shown to improve energy per part while preserving throughput and outperforms the strongest model‑free baseline (DQN variants tuned for this environment). The paper argues that full observation reconstruction is not necessary for effective control once the EFE‑trained policy stabilizes.

(b) This paper presents an interesting link between AIF, MBRL and world models. By placing the actor inside the generative model and optimizing it via EFE gradients, MCTS can be avoided and can be scaled to continuous action spaces. The approach has long horizon capabilities. The conceptual framework is quite interesting and it deserves further investigation.

(c) The biggest limitation is the narrow empirical scope. Only one environment evaluation is shown and there are no head-to-head comparisons against recent world-model RL approach (e.g. Dreamer) or strong AIF baselines. I agree with the authors and for wider impact, this line of work needs to expand its experimental scope to include a larger set of environments used in the RL community. Without this leap, it is difficult to get a strong consensus for acceptance. There are writing and consistency issues in the paper as highlighted during the rebuttal period.

(d) This paper got borderline scores from reviewers and the primary reason is due to the narrow empirical scope as described in the above section. Once this is addressed, I think this paper and line of work can have a big impact on the RL and world modeling communities.

(e) Two reviewers recommend borderline accept (oAWD and sddb), while one remains borderline reject (mpAY) chiefly due to the limited breadth of evaluation. During the rebuttal many of the other concerns were addressed but this core question remain unresolved. mpAY summaries the core issue succinctly - “the main criticism remains in that for Neurips, a proof of concept on a very limited domain is not sufficient to show that this is a useful, generalizable result. While it may indeed be, this needs to be shown by comparing with a range of SOTA algorithms on a range of problems, not necessarily to show superiority, but to show that it is indeed a useful framework. As it currently stands, I don't believe that this has been shown strongly enough for Neurips main track.”